# Mindfulness and Defense Mechanisms as Explicit and Implicit Emotion Regulation Strategies against Psychological Distress during Massive Catastrophic Events

**DOI:** 10.3390/ijerph191912690

**Published:** 2022-10-04

**Authors:** Mariagrazia Di Giuseppe, Graziella Orrù, Angelo Gemignani, Rebecca Ciacchini, Mario Miniati, Ciro Conversano

**Affiliations:** 1Department of Surgical, Medical and Molecular Pathology, Critical and Care Medicine, University of Pisa, 56126 Pisa, Italy; 2Department of History, Culture and Society, University of Rome Tor Vergata, 00133 Rome, Italy

**Keywords:** emotion regulation, mindfulness, defense mechanisms, stress, COVID-19

## Abstract

Emotion regulation is an important aspect of psychological functioning that influences subjective experience and moderates emotional responses throughout the lifetime. Adaptive responses to stressful life events depend on the positive interaction between explicit and implicit emotion regulation strategies, such as mindfulness and defense mechanisms. This study demonstrates how these emotion regulation strategies predict psychological health during the early phase of the COVID-19 pandemic. A convenience sample of 6385 subjects, recruited via snowball sampling on various social media platforms, responded to an online survey assessing psychological reaction to social restrictions imposed to limit the spread of COVID-19 in Italy. Psychological distress, post-traumatic stress symptoms, mindfulness, and defense mechanisms were assessed using SCL-90, IES-R, MAAS, and DMRS-30-SR, respectively. Higher mindfulness was significantly associated with higher overall defensive maturity and a greater use of high-adaptive defenses (*p* < 0.0001). Both mindfulness and defense mechanisms acted as good predictors of psychological health (R^2^ = 0.541) and posttraumatic symptoms (R^2^ = 0.332), confirming the role of emotion regulation in protecting against maladaptive responses to stressful situations.

## 1. Introduction

The role of emotion regulation as a central aspect of mental health has been widely demonstrated and is still of increasing interest in clinical psychology because of its impact on chronic diseases [1,2,3,4,5,6]. Emotion regulation is defined as the ability to influence subjective experience and expression of emotions and includes all conscious and unconscious strategies activated to moderate emotional responses [7]. The conscious effort to control and change an emotional reaction is defined as explicit emotion regulation [8]. It generally includes cognitive reappraisal and expressive suppression. Recent neurobiological models propose that mindfulness may lead to changes in self-processing through the development of self-awareness, self-regulation, and self-transcendence, which may reflect modulation in neurocognitive networks related to intention and motivation, attention, and emotion regulation [9,10]. Implicit emotion regulation operates outside of conscious awareness and encompasses unintentional and automatic processes, such as defense mechanisms and somatization. Implicit emotion regulation may be even more important than explicit strategies in maintaining psychological health [11,12]. According to the dual-process framework of emotion regulation, adaptive responses depend on the positive interaction between both explicit and implicit emotion regulation processes [13].

Mindfulness is defined as the experience of awareness activated by purposely paying attention to what occurs in the present with a non-judgmental attitude [14]. It is considered a multi-facet construct, involving attention, awareness, and an open-minded acceptance of the present moment, namely an open and accepting attitude, and an ability to take a step back from one’s experience without immediately reacting to it [15]. Psychological interventions based on improving mindfulness disposition have been found to be highly effective in reducing mental health symptoms, such as stress, anxiety, and depression [16], and in enhancing positive coping during the COVID-19 pandemic. For example, Zhu et al. [17], described lower levels of pandemic-related distress in mindfulness practitioners compared to non-practitioners. In a study conducted during the early days of the COVID-19 pandemic, Conversano et al. [18] found that dispositional mindfulness was associated with mental health as the best predictor of lower psychological distress among a number of socio-demographics and psychological factors related to COVID-19; these data are also supported by the application of machine learning [19,20]. Similarly, Kock et al. [21] investigated the impact of specific mindfulness facets on adolescents’ psychological functioning during the COVID-19 pandemic. They found that decentering a mindfulness facet was associated with decreased worry and stress, improved mental health, enhanced quality of life, and increased social connectedness with others.

Defense mechanisms are automatic psychological strategies that subjects might use to protect themselves from anxiety produced by internal conflicts and external stressors. Usually, defense mechanisms are hierarchically organized into a continuum of maturity and adaptiveness [22], with mature defenses sharing several common aspects with so-called coping strategies [23]. A wide body of research has already demonstrated that defense mechanisms might be significant predictors of physical and mental health [24,25,26,27,28], especially under stressful conditions, such as the COVID-19 pandemic [29,30,31]. According to Di Giuseppe et al. [32], greater levels of psychological distress, post-traumatic symptoms, depression, and anxiety might be associated with lower overall defensive functioning (ODF). Moreover, each increased unit of ODF might decrease the chances of developing post-traumatic stress symptoms by 71%. The relationship between life satisfaction and perceived stress could be partially mediated by approach coping, positive attitude, and mature defenses, as confirmed by Gori et al. [33]. Similarly, Di Giuseppe et al. [34] found that mature defensive functioning in healthcare professionals working during the COVID-19 pandemic was associated with resilience and personal accomplishment, while neurotic and immature defenses were related to perceived stress and burnout, known to worsen mental health in various conditions [35,36,37]. Furthermore, Aafjes-van Doorn et al. [38] showed that therapists’ lower defensive functioning was related to higher levels of vicarious trauma and professional doubt, thus enhancing the importance of clinicians’ emotion regulation in the therapeutic process [39,40,41,42,43].

Our study aimed to analyze the role of mindfulness and defense mechanisms in protecting against psychological distress during catastrophic events, such as the COVID-19 pandemic. Although both mindfulness and defense mechanisms may contribute to emotion regulation, most research has been focused on one aspect at time, and has separately investigated the role of explicit and implicit components of emotion regulation in stress management during catastrophic events. To the best of our knowledge, this is the first study investigating both mindfulness and defense mechanisms in a general population sample, namely Italian residents under lockdown restrictions imposed during the first wave of the COVID-19 pandemic. In more detail, we were interested in: (H1) the associations between explicit and implicit emotion regulation strategies as mindfulness and defense mechanisms; (H2) the role of mindfulness and defense mechanisms in protecting from psychological distress and post-traumatic symptoms during the initial spread of COVID-19; and (H3) the role of emotion regulation in predicting specific psychiatric symptoms. We hypothesized that different explicit and implicit emotion regulation strategies might be reciprocally associated, and that emotion regulation might predict lower psychological distress in a general population sample dealing with an unexpected and terrifying situation, such as the first wave of the COVID-19 pandemic.

## 2. Methods

### 2.1. Participants and Procedures

A convenience sample of 6385 Italian residents responded to an online survey launched on 13 March 2020 and closed after 2 weeks, on 26 March 2020. This time span corresponded to the very early days of the spread of COVID-19 in Italy, when the Italian government imposed the first national lockdown to limit the pandemic. Subjects were recruited using snowball sampling on various social media platforms (i.e., Facebook, Instagram, WhatsApp, etc.), informed about the purpose of the study, and asked to give their approval for the use of their personal data. Participants provided socio-demographic and COVID-related information, including the presence/absence of positive cases or deaths among relatives and friends and whether they had moved to another location as a result of the pandemic [18,32]. Exclusion criteria were (1) being less than 18 years of age, and (2) not signing the online consent form. The study procedure was reviewed and approved by the local [omitted for peer review] Institutional Review Board.

### 2.2. Measures

The online survey included the following instruments in addition to socio-demographic information: the Symptoms Checklist-90 (SCL-90) [44], for the assessment of general psychiatric symptoms; the Impact of Event Scale-Revised (IES-R) [45], for post-traumatic stress symptoms; the Mindfulness Attention Awareness Scale (MAAS) [46], for mindfulness disposition; the Defense Mechanisms Rating Scales-Self-Report-30 (DMRS-SR-30) [47], for the assessment of defense mechanisms. All questionnaires were validated in Italian and their internal consistency resulted in Cronbach’s alphas of 0.92, 0.88, 0.87, and 0.890, respectively.

The Symptoms Checklist-90 is a 90-item self-report, assessing psychopathological and somatic symptoms on a 5-point Likert scale. This measure provides scores for a Global Severity Index (GSI) and nine psychiatric symptoms: Somatization (SOM); Obsessive-Compulsive (O-C); Interpersonal Sensitivity (I-S); Depression (DEP); Anxiety (ANX); Hostility (HOS); Phobic Anxiety (PHOB); Paranoid Ideation (PAR); Psychoticism (PSY). Psychometric properties of the scale are widely documented [44,48]. Internal consistency for the present study was 0.97.

The Impact of Event Scale-Revised is a 22-item self-report assessing an overall index of post-traumatic stress symptoms and three subscales reflecting the specific symptoms of intrusion, avoidance, and hyperarousal. The IES-R is often used as a screening instrument for post-traumatic stress disorder and has good psychometric properties [49,50], with an internal reliability of 0.94 in the present study.

The Mindfulness Attention Awareness Scale is a 15-item self-report assessing mindfulness disposition on a 6-point Likert scale. This scale measures the frequency of open and receptive attention to, and awareness of, ongoing events and experiences. MAAS items are presented as negative descriptions of mindfulness, meaning that higher scores indicate lower mindfulness. Adequate test–retest reliability, and convergent and discriminate validity have been demonstrated for this scale [51,52]. Internal consistency for the present study was 0.87.

The Defense Mechanisms Rating Scales-Self-Report-30 (DMRS-SR-30) [31] is a 30-item self-report instrument assessing defense mechanisms on a 5-point Likert scale. It provides scores for Overall Defensive Functioning (ODF), 3 defensive factors, 7 hierarchically ordered defense levels, and 28 defense mechanisms. This measure has good psychometric properties; very good reliability and criterion; and concurrent, convergent and discriminant validity [47,53]. Internal consistency for the present study was 0.89.

### 2.3. Statistical Analyses

Descriptive data are presented as means, standard deviations, and 95% confidence intervals. The Anderson–Darling test and Normal P-P plot were used to verify the normality of distributions. Pearson correlation coefficients were used to calculate associations between explicit and implicit emotion regulation strategies. Finally, multivariate analysis was used to test mindfulness and defense mechanisms as moderators of psychological symptoms.

## 3. Results

### 3.1. Sample’s Psychological Characteristics

Participants were prevalently females (N = 4797; 75,1%); adults younger than 40 years (N = 3465; 54,3%); living with close relatives (N = 4486; 70,3%); and without children (N = 3779; 59,2%). Average scores for all psychological variables analyzed in the study are summarized in Table 1. Mindfulness (M = 1.12; SD = 0.65) and defensive functioning (M = 5.60; SD = 0.73) mean scores were around normative values for healthy individuals. Mean scores of psychological distress and psychiatric symptoms, including post-traumatic stress symptoms, fell within the clinical range [54,55,56].

### 3.2. Associations between Mindfulness and Defense Mechanisms

Table 2 summarizes correlations between mindfulness dispositions calculated with the MAAS and defense indexes calculated with the DMRS-SR-30. As expected, high significant negative correlations were found between MAAS and both ODF and Factor 1 (*r* = −0.497 and *r* = −0.540, respectively; *p* < 0.0001), suggesting that higher mindfulness was associated with higher defensive maturity and a greater use of mature defenses. Conversely, high significant positive correlations were found between MAAS and Factors 2 and 3, suggesting that lower mindfulness might be associated with greater use of mental inhibition and avoidance defenses as well as immature–depressive defenses.

### 3.3. Moderating the Role of Emotion Regulation in Protecting against Psychological Problems

A series of multivariate linear regression analyses were performed to assess the moderating role of mindfulness and defense mechanisms in determining psychological conditions, such as overall psychological distress, post-traumatic stress, and other psychiatric symptoms.

Table 3 summarizes the results of multivariate linear regressions for explicit and implicit emotion regulation predicting psychological distress, calculated according to the GSI. Independent variables included in the model were: (1) MAAS, a negative index of mindfulness; (2) ODF, a positive index of defensive maturity; (3) Factor 1, including all mature defense mechanisms; (4) Factor 2, including all mental inhibition and avoidance defense mechanisms; and (5) Factor 3, including all immature–depressive defense mechanisms. All variables acted as good predictors of GSI, explaining 54.1% of the variance. For each one-unit increase in mindfulness and overall defensive maturity the GSI decreased by 0.35 and 0.22, respectively.

Similar results emerged from the multivariate linear regressions for explicit and implicit emotion regulation predicting post-traumatic stress symptoms, calculated as IES-R and displayed in Table 4. The same five variables entered the model and all of them were significant predictors of IES-R, explaining 33.2% of the variance. As for the GSI, mindfulness and defensive functioning were the best predictors of the dependent variable and there was an increase of almost 10 units in the IES-R for each one-unit decrease in MAAS and ODF.

Multivariate linear regression analyses were also performed on several psychiatric symptoms calculated as subscales of the SCL-90 and IES-R. The same model was replicated for each symptom, as reported in Table 5. All independent variables acted as good predictors of all psychiatric symptoms, with the exception of paranoid ideation (SCL-90-PAR) and avoidance (IES-R-Avoidance), which were predicted by mindfulness, defensive functioning and one or two defensive factors instead of all of them. The variance explained by emotion regulation factors on psychiatric symptoms was high, ranging from 21.2% to 49.8%.

## 4. Discussion

Our study demonstrated the reciprocal contribution of mindfulness and defense mechanisms in helping the management of high stress levels related to the COVID-19 pandemic. Explicit and implicit emotion regulation strategies were both investigated using well-validated measures. In particular, we used the questionnaire based on the gold-standard DMRS theory [47,57] to assess the whole hierarchy of defensive strategies. This methodological accuracy allowed for an in-depth qualitative and quantitative exploration of convergent and divergent aspects of explicit and implicit emotion regulation, usually studied separately [58,59]. The main aim of this study was to explore to what extent both dispositional mindfulness and defensive functioning might influence mental health in specific stressful conditions (such as the COVID-19 pandemic).

With regard to the first hypothesis, the results confirmed our expectations, namely that dispositional mindfulness would be associated with high-adaptive defense mechanisms. Significant high correlations were found between MAAS and DMRS-SR-30 scores, and they were in the positive relationship. The strongest relationship was between mindfulness and mature defenses, confirming a previous finding from Di Giuseppe et al. [23]. The hierarchy of defense mechanisms [22] is a comprehensive description of a continuum from maladaptive to adaptive defensive strategies, where defenses higher in the hierarchy (i.e., mature defenses) might share some overlapping functions with explicit emotion regulation strategies. For instance, the awareness of what the individual might experience in the present moment, which is one fundamental aspect of dispositional mindfulness, was also an important component of the mature defense self-observation, in which “*the individual deals with emotional conflicts or conflictual ideas by reflecting on his or her own thoughts, feelings, motivation, and behavior. The person can see himself as others see him in interpersonal situations, and as a result is better able to understand other people’s reactions to him or her*” [22]. Conversely, lack of attention, awareness, and open-minded acceptance of the present moment, typical of people with low mindfulness, were common in individuals who frequently might revert to immature defenses.

According to the second hypothesis, mindfulness and defense mechanisms would protect from psychological distress and post-traumatic symptoms during the initial spread of the COVID-19 pandemic. Study findings confirmed the moderating role of both explicit and implicit emotion regulation strategies. Results from multivariate linear regressions showed that dispositional mindfulness and defensive functioning explained up to 54% and 33% of the variance on symptom severity and post-traumatic stress, respectively. Looking at the regression coefficients, the two global indexes of MAAS and ODF were the main contributors to the explained variance.

We should also note that the DMRS-SR-30 [60], a comprehensive self-report instrument based on the gold-standard theory of defenses, was administered during this study, providing several levels of scoring and an overall index of defensive maturity [47,53]. Most of the previously available measures for defensive assessment did not allow for a global evaluation of defensive maturity and were often limited to the assessment of a few defense mechanisms, instead of the full hierarchy of defenses. We believe that these methodological limitations inevitably led to a systematic bias, which could be prevented by using more extensive measures, such as those based on the DMRS [61]. Future studies are needed to further test the validity of the measure according to Modern Test Theory methods [62].

Our last hypothesis was that emotion regulation would predict specific psychiatric symptoms. We found significant results from all multivariate linear regressions carried out on SCL-90 and IES-R subscales. Mindfulness and defense mechanisms together were significant predictors of 12 psychiatric symptoms, with R^2^ ranging from 0.212 to 0.498. The high values of variance explained by these two variables on all analyzed symptoms confirmed the key role of emotion regulation in the etiopathogenesis of psychological problems. Interestingly, the independent variables that were entered in the model and replicated for each sign were all significant predictors of psychiatric symptoms, except for paranoid ideation and avoidance. The two global indexes of mindfulness and defensive functioning were the best predictors for these two regression models, while immature defenses was the only defensive factor predicting paranoid ideation and the only non-significant factor predicting avoidance. We interpreted these results as an indication of the potential impact of specific defense mechanisms in symptom formation.

Our study had some limitations. Results might be biased by the snowball sampling method because of the introduction of uncontrolled variables. Moreover, the causal relationships between tested variables could not be determined, giving the cross-sectional research design. Furthermore, the use of self-reported measures might have determined response biases typical of self-assessment of psychological variables. Despite these limitations, our study demonstrated associations between different emotion regulation strategies and highlighted the key role of mindfulness and defense mechanisms in moderating the experience of COVID-19-related distress.

## 5. Conclusions

The impact of emotion regulation in mediating an individual’s adjustment to traumatic experiences is remarkable. The adaptive function of emotion regulation becomes of great importance in massive catastrophic events such as the experience of the COVID-19 pandemic. As an isolated attempt at studying simultaneously explicit and implicit emotion regulation, this study aimed to pave the way for new research on the interplay between psychological resources and stressful life conditions in both general and clinical populations [63,64,65]. The systematic investigation of emotion regulation with appropriate instruments could be an important support in the early detection of vulnerable individuals at risk of developing several psychopathologies. Future research should address the need to validate emotion regulation-based psychological intervention, enhancing resilience to and preventing the development of mental disorders [66,67].

## Figures and Tables

**Table 1 ijerph-19-12690-t001:** Descriptive statistics of responders’ psychological characteristics (N = 6385).

Psychological Distress (SCL-90)	Min	Max	Mean	SD	95% i0.c0.	
					Lower	Upper
Global Severity Index (GSI)	0.00	30.60	0.73	0.53	0.00	30.60
SCL-90 SOM	0.00	30.92	0.62	0.60	0.00	30.92
SCL-90 O-C	0.00	30.70	0.87	0.68	0.00	30.70
SCL-90 INT	0.00	40.00	0.61	0.59	0.00	40.00
SCL-90 DEP	0.00	30.92	0.98	0.75	0.00	30.92
SCL-90 ANX	0.00	30.70	0.86	0.69	0.00	30.70
SCL-90 HOS	0.00	40.00	0.65	0.61	0.00	40.00
SCL-90 PHOB	0.00	30.86	0.48	0.55	0.00	30.86
SCL-90 PAR	0.00	40.00	0.72	0.66	0.00	40.00
SCL-90 PSY	0.00	40.00	0.51	0.52	0.00	40.00
SCL-90 SLEEP	0.00	40.00	10.08	0.98	0.00	40.00
**Post-traumatic stress symptoms (IES-R)**						
IES-R	0.00	860.00	250.25	160.19	0.00	860.00
Intrusion	0.00	320.00	80.32	60.28	0.00	320.00
Avoidance	0.00	320.00	90.73	60.62	0.00	320.00
Hyperarousal	0.00	240.00	70.20	40.87	0.00	240.00
**Mindfulness (MAAS)**						
MAAS ^a^	0.00	50.00	10.12	0.65	0.00	40.00
**Defense Mechanisms (DMRS-SR-30)**						
Overall Defensive Functioning (ODF)	10.00	70.00	50.60	0.73	10.00	70.00
Factor 1: Mature	0.00	1000.00	550.66	190.62	0.00	1000.00
Factor 2: Mental inhibition and Avoidance	0.00	1000.00	240.89	120.10	0.00	1000.00
Factor 3: Immature	0.00	1000.00	170.80	130.36	0.00	1000.00

Note: ^a^ Lower MAAS score indicates higher mindfulness.

**Table 2 ijerph-19-12690-t002:** Pearson correlations between mindfulness disposition and defense mechanisms (N = 6385).

	Mindfulness (MAAS) ^a^	
Defense mechanisms (DMRS-SR-30)	*r*	*p*
Overall Defensive Functioning (ODF)	−0.497	<0.0001
Factor 1: Mature	−0.540	<0.0001
Factor 2: Mental inhibition and Avoidance	0.385	<0.0001
Factor 3: Immature	0.444	<0.0001

Note: ^a^ Lower MAAS score indicates higher mindfulness.

**Table 3 ijerph-19-12690-t003:** Multivariate linear regressions for mindfulness and defense mechanisms predicting psychological distress (N = 6385).

	Overall Psychological Distress (GSI)						
	B	SE	t	*p*	F	*p*	Adjusted R^2^
MODEL					1507.82	<0.0001	0.541
ODF	−0.220	0.028	−7.874	<0.0001			
Factor 1	−0.016	0.002	10.253	<0.0001			
Factor 2	0.017	0.001	11.151	<0.0001			
Factor 3	0.022	0.002	13.802	<0.0001			
MAAS	0.353	0.008	42.234	<0.0001			

Note: Independent variables entered in the multivariate linear regression model were: ODF, Factor 1, Factor 2, Factor 3, and MAAS. ^a^ ODF is the acronym of overall defensive functioning; Factor 1 includes mature defenses; Factor 2 includes mental inhibition and avoidance defenses; Factor 3 includes immature-depressive defenses.

**Table 4 ijerph-19-12690-t004:** Multivariate linear regressions for mindfulness and defense mechanisms predicting post-traumatic stress symptoms (N = 6385).

	Post-Traumatic Stress Symptoms (IES-R)						
	B	SE	t	*p*	F	*p*	Adjusted R^2^
MODEL					635.02	<0.0001	0.332
ODF ^a^	−9.569	1.02	−9.385	<0.0001			
Factor 1	−0.500	0.059	8.509	<0.0001			
Factor 2	0.489	0.054	8.992	<0.0001			
Factor 3	0.265	0.057	4.606	<0.0001			
MAAS	10.11	0.305	33.168	<0.0001			

Note: Independent variables entered in the multivariate linear regression model were: ODF, Factor 1, Factor 2, Factor 3, and MAAS. ^a^ ODF is the acronym of overall defensive functioning; Factor 1 includes mature defenses; Factor 2 includes mental inhibition and avoidance defenses; Factor 3 includes immature-depressive defenses.

**Table 5 ijerph-19-12690-t005:** Multivariate linear regressions for mindfulness and defense mechanisms predicting psychiatric symptoms (N = 6385).

	Model Summary		
Symptoms	F	*p*	F
Somatization (SCL-90-SOM)	4420.1	<0.0001	4420.1
Obsessive Compulsive (SCL-90-O-C)	11,530.10	<0.0001	11,530.10
Interpersonal Sensitivity (SCL-90-I-S)	12,670.34	<0.0001	12,670.34
Depression (SCL-90-DEP)	9560.68	<0.0001	9560.68
Anxiety (SCL-90-ANX)	7020.48	<0.0001	7020.48
Hostility (SCL-90-HOS)	9000.31	<0.0001	9000.31
Phobic Anxiety (SCL-90-PHOB)	3450.05	<0.0001	3450.05
Paranoid Ideation (SCL-90-PAR) ^a^	11,300.15	<0.0001	11,300.15
Psychoticism (SCL-90-PSY)	2880.28	<0.0001	2880.28
Intrusion (IES-R-Intrusion)	4010.37	<0.0001	4010.37
Avoidance (IES-R-Avoidance) ^b^	4270.83	<0.0001	4270.83
Hyperarousal (IES-R-Hyperarousal)	7770.04	<0.0001	7770.04

Note: Independent variables entered in the multivariate linear regression model were: ODF, Factor 1, Factor 2, Factor 3, and MAAS. ^a^ Regression coefficients for Factor 1 and Factor 2 were non-significant; ^b^ regression coefficient for Factor 3 was non-significant.

## Data Availability

The data presented in this study are available on request from the corresponding author. The data are not publicly available due to ethical restrictions.

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
