# Peer review of "Mindfulness and Defense Mechanisms as Explicit and Implicit Emotion Regulation Strategies against Psychological Distress during Massive Catastrophic Events"

_ijerph, 2022, doi:10.3390/ijerph191912690_

Round 1

Author Response

We thank the reviewer for his/her careful and interesting comments. We appreciated this review, and we did our best to meet the reviewer’s request of revision. However, we disagree with some of the comments and in these cases we preferred to leave the manuscript in the present form. We explained our reasons below:

 General Comments:

Reviewer: This study examined how implicit and explicit forms of emotion regulation affected coping outcomes during the recent COVID pandemic. In my view, this topic is pertinent to the Journal’s readership only if it relates clearly and precisely relates to the environment in a biopsychosocial sense.

This submission seems to touch on this, but there are too many ambiguities to draw meaningful conclusions (much less clinical or public policy implications) from the data here. I would urge the authors to treat this as a pilot study and redo the procedure with improved protocols. Of course, this might be impossible unless active lockdowns are still occurring.

Authors: We understand the reviewer’s comment, that suggested to treat this study as a pilot study. However, as he/she properly mentioned, this is impossible since the first months of lockdown for COVID-19 in Italy has been an unique experience that we would never be able to replicate in any future study. Thus, the present study is actually very far from the concept that we authors have of a pilot study.

Specific Comments:

Reviewer: Abstract: Very vague descriptions that require more precision to give a meaningful summary. For instance, readers are neither given key details context about the respondent pool and the findings, nor specific effect sizes for the apparent findings.

Authors: We revised the abstract according with the reviewer comments and added relevant results in order to stress the impact of our findings.

Reviewer: Introduction: There are many grammatical errors here and elsewhere throughout the paper. The draft thus requires thorough proofreading and editing. More critically, the authors do not make a compelling case for why they specifically focused on implicit/explicit emotion regulation and defense mechanisms in this study. Their Introduction nicely covers relevant and timely literature, but they seem to have a priori selected target variables and used certain variables to justify them post hoc. It would be more impartial to conduct a general lit review that spoke to “established coping strategies” for  “illness,  crisis, and/or isolation” and then used the outcomes to identify the independent and dependent variables.

Authors: Due to revision submission deadline, we were not able to get proofreading on time. We will provide English revision later, during the proof correction phase, if accepted. In the introduction, we gave not only a general explanation of what it is meant as emotional regulation strategies, but also how this psychological construct includes different conscious and unconscious processes as mindfulness and defense mechanisms. Later in the section, we defined these constructs with numerous lit references in order to introduce our research hypotheses. We did not select a priori target variables, we just decided to investigate defense mechanisms and mindfulness and not include coping strategies. For this reason we reported in the introduction only findings related to the investigated variables.

Reviewer: Method - Participants: It should be emphasized that this was a convenience sample. Moreover, the time frame for the survey was arguably not associated with the most stringent or stressful periods of lockdowns. Rather, the early period was often associated with social approval and compliance on a global scale vs after an extensive period of isolation. Were the respondents compensated in any way for their participation? Were the respondents screened or otherwise demarcated as to whether they were or had been infected or were currently dealing with COVID symptoms? This seems an obvious but enormously important confounding variable that perhaps was overlooked somehow.

Authors: In the introduction We must disagree with the reviewer’s comment that “the early period was often associated with social approval and compliance on a global scale vs after an extensive period of isolation”. In Italy, that was one of the very first country affected by the Coronavirus, the initial phase of the spreading was lived as extremely stressfull, as we and other Italian colleagues demonstrated in several previous studies. The first lockdown was extremely strict, long and unusual. Most people could not work, except frontline workers. None was able to get access to usual routines, we were allowed to have a walk only 30 minutes around our house once a day. We might mention many other restrictions related to the first lockdown to sustain this thesis, but we believe that this is not necessary since research already demonstrated that the first lockdown in Italy was the harder one for psychological health.

Reviewer: Method – _Measures: The authors need to better justify why these particular questionnaires were used and give specific psychometric descriptions. As it is, the descriptions are not consistently detailed. Plus, it is unclear whether the cited reliabilities correspond to past research or the present sample. They should be reported for the present sample. Most critically, it neither seems that the questionnaires were administered in counterbalanced order, nor that any of the questionnaires were developed and validated via Modern Test Theory methods (cf. Lange, 2017). Thus, there is no guarantee that they provide unbiased and interval-level measurements. As a result, non-parametric statistics really are most appropriate here. Finally, we do not know the instruction set given to the respondents, thus, there is a fatally ambiguous context for the respondents when they completed their questionnaires. Specifically, how do the authors know to what stressor(s) the scores on the various measures correspond to? For instance, respondents could be indicating their (a) fear or anxiety of being infected, (b) fear or anxiety about the consequences of their current infection, (c) psychophysical responses to dealing with an active infection, (d) psychological feelings associated with a recovery period, (d) psychological effects of isolation, or (e) any combination of the former options. So, unless the respondents were given a clear and consistent context for answering the questionnaires, we do not know to what stimulus or stimuli  their responses correspond. It’s the same basic issue as using survey statements that are compound sentences.

Authors: In this study we used very well-known measures validated in many languages. For this reason we did not report psychometric properties but we do reported references for validation studies for each measure. The only new measure that we used in this research, which was developed by some of the authors, is also well validated and we provided references for validation studies as well. Because of the strong psychometric properties of these measures and the large sample that we collected, we believe that we used appropriate level of statistical analyses.

Regarding the reviewer’s comment that “how do the authors know to what stressor(s) the scores on the various measures correspond to? For instance, respondents could be indicating their (a) fear or anxiety of being infected, (b) fear or anxiety about the consequences of their current infection, (c) psychophysical responses to dealing with an active infection, (d) psychological feelings associated with a recovery period, (d) psychological effects of isolation, or (e) any combination of the former options.”, this was not the goal of our reseach. We investigated some of these issues we previous studies.

Reviewer: Results: The authors place great emphasis on statistical significance over replication. The large sample size amplifies the risk for Type I errors, especially when there are no controls or corrections for multiple observations. The authors would do better to utilize a split-sample approach here. This involves randomly dividing the data into (a) Training and (b) Validation samples. Thus, statistical effects are examined in Sample A and any significant findings are immediately replicated (or attempted) in Sample B. Moreover, regressions are rather simplistic here. Path analysis or structural equation modeling would be more interesting and comprehensive at building and testing competing process models that involve all the variables simultaneously. Lastly, why were “age or gender” not assessed in the analyses?

Authors: According with the reviewer’s comment, we decrease the emphasis on statistical significance over replication. We did not included age and gender in the analyses since we already widely treated demographic variables in previous papers on the topic. We preferred to avoid possible overlaps and instead focus on emotion regulation strategies 

Reviewer: Discussion: The authors should greatly temper confidence in their results and conclusions given that (a) there were potentially serious methodological shortcomings that preclude firm conclusions about the context for the results, (b) their “gold-standard measures” are not convincing from the actual psychometric gold standard of Modern Test Theory, and (c) no apparent controls for multiple observations with large sample sizes (much less replication of the results). In my opinion, these shortcomings negate any clear interpretation of the results and thus their clinical or theoretical import is virtually nullified.

Authors: The “gold-standard” is related to the theoretical frame for assessing defense mechanisms, the well-known DMRS theory. The measure that we used in this study is based on this remarkable theory and methodology, which has been demonstrated in more that 30 years of research in defense mechanisms. Although we believe in the high significance of our findings despite the limitation of the research design, we slow down our temper and highlighted the limits of the study as suggested by the reviewer.

Reviewer 2 Report

Thank you for the opportunity to review this interesting and well-written paper. I only have a few minor comments:

1. Table 1: I think it would help to add the range of possible values and the min/max values for each tool

2. You mention moderation in your hypotheses, but I did not see any specific moderation analysis in the results section. Please clarify. 

Author Response

We thank the reviewer for his/her positive comments on our study. We appreciated reviewer’s suggestions and provided all requested changes in Table 1 and in the formulation of hypotheses.

Reviewer 3 Report

The Authors present an original and interesting research titled "Mindfulness and defence mechanism as explicit and implicit emotion regulation strategies against psychological distress during massive catastrophic events".  The analyse the role of both mindfulness and defence mechanisms as protectors against psychological distress during catastrophic events, such as Sars-Cov-19 pandemic, offering a very new interesting contribution to the field. I found a high quality of presentation and a very good scientific soundness. The introduction provides adequate background; the research design is very smart and well described; results are well presented, sufficiently reporting conclusions. Even the limits are quite approached. Definitely, authors demonstrated associations between different emotion regulation strategies, highlighting the key role of mindfulness and defence mechanisms in moderating the Covid-19 related distress.

Author Response

We sincerely thank the reviewer for his/her positive comments. We are grateful for his/her appreciation of the study in the present form.

Round 2

Reviewer 1 Report

The authors made some good revisions to the original draft, but still there are (a) too many ambiguities with the procedure and (b) poor controls for multiple observations. In fact, the authors ignored my specific suggestions that would have alleviated some of these concerns. First, readers should be given the internal reliabilities of the measures in the present sample; it is not enough merely to cite studies about the psychometric properties of these tools. Second, these tools were neither constructed nor validated with Modern Test Theory methods, so the "scores" do not provide interval-level measurements. As a result, parametric statistics are inappropriate. Third, the sample size is sufficiently large to analyze the data via a split-sample approach that emphasizes the critical issue of replicable results in the social sciences. Statistical significance itself is not compelling anymore. Lastly, the confounds with the sample (and directions given to the respondents) have not been satisfactorily addressed. Plus, no critical contextual data were collected in tandem with stress/symptoms levels so it remains unclear to what stimulus/stimuli the scores on the distress measures were associated.

Overall, the authors certainly improved the Abstract and Introduction sections, but the methodology was weak and lacked critical context, and the analysis should have applied non-parametric statistics and strived for replication over mere significance.

Author Response

We thank the reviewer for his/her helpful comments. We appreciated his/her suggestions, which helped our reflection about future research. We further improved our manuscript by mentioning (1) internal reliability of the measures in the present study and (2) directions given to the respondents. We also added (3) references of previous papers where we provided all information.

Although we agree with reviewer’s suggestion of applying Modern Test Theory to test validation, that we will certainly consider in future validation studies, we also notice that most of the scientific articles published in high-standard journal used measures constructed and validated outside this paradigm. And authors did use parametric test. Aware of this limitation, we preferred to keep our original statistical analyses and (4) add in the discussion a sentence highlighting the need of test validations with the Modern Test Theory. About the split-sample approach suggested, we think it could be bypassed since it was not the focus of this paper. We planned a deeper analysys of the sample in future research.

Thank you again for all comments and suggestions.

Best Regards,

All authors